# Lipid Droplet-Associated Proteins Perilipin 1 and 2: Molecular Markers of Steatosis and Microvesicular Steatotic Foci in Chronic Hepatitis C

**DOI:** 10.3390/ijms232415456

**Published:** 2022-12-07

**Authors:** Selina Schelbert, Mario Schindeldecker, Uta Drebber, Hagen Roland Witzel, Arndt Weinmann, Volker Dries, Peter Schirmacher, Wilfried Roth, Beate Katharina Straub

**Affiliations:** 1Institute of Pathology, University Medical Center Mainz, 55131 Mainz, Germany; 2Institute of Pathology, University Hospital Wuerzburg, 97080 Wuerzburg, Germany; 3Institute of Pathology, University Clinic Cologne, 50931 Cologne, Germany; 4Department of Internal Medicine, University Medical Center, 55131 Mainz, Germany; 5Institute of Pathology, University Medical Center Heidelberg, 69120 Heidelberg, Germany

**Keywords:** PAT proteins, hepatitis C virus (HCV), hepatitis B virus (HBV), non-alcoholic steatohepatitis (NASH), focal fatty change

## Abstract

Chronic infection with hepatitis C (HCV) is a major risk factor in the development of cirrhosis and hepatocellular carcinoma. Lipid metabolism plays a major role in the replication and deposition of HCV at lipid droplets (LDs). We have demonstrated the importance of LD-associated proteins of the perilipin family in steatotic liver diseases. Using a large collection of 231 human liver biopsies with HCV, perilipins 1 and 2 have been localized to LDs of hepatocytes that correlate with the degree of steatosis and specific HCV genotypes, but not significantly with the HCV viral load. Perilipin 1- and 2-positive microvesicular steatotic foci were observed in 36% of HCV liver biopsies, and also in chronic hepatitis B, autoimmune hepatitis and mildly steatotic or normal livers, but less or none were observed in normal livers of younger patients. Microvesicular steatotic foci did not frequently overlap with glycogenotic/clear cell foci as determined by PAS stain in serial sections. Steatotic foci were detected in all liver zones with slight architectural disarrays, as demonstrated by immunohistochemical glutamine synthetase staining of zone three, but without elevated Ki67-proliferation rates. In conclusion, microvesicular steatotic foci are frequently found in chronic viral hepatitis, but the clinical significance of these foci is so far not clear.

## 1. Introduction

The pathologic accumulation of micro- or macrovesicular lipid droplets (LDs) in hepatocytes, plays an important role in steatotic liver diseases including alcoholic and non-alcoholic liver diseases and their forms alcoholic/non-alcoholic fatty liver disease and steatohepatitis (ALD/AFLD/ASH, and NAFLD/NASH) [1,2] as well as chronic hepatitis C [3,4]. Steatotic liver diseases account for the majority of chronic liver diseases worldwide (for USA see, e.g., [5]). The global prevalence of NAFLD, the main cause of mild liver disease, is estimated at 24% with the highest rates in South America and the Middle East, due to the pandemic increase in obesity [6]. ALD, however, remains to be the main cause of liver-related mortality cases worldwide [7,8]. In addition to classic steatotic liver diseases, steatosis also plays a major role in chronic viral hepatitis, especially in patients infected with HCV. In 2015, 71 million people were infected with HCV, and 1.34 million people died as a consequence of viral hepatitis in the same year [9]. Importantly, comorbidity induced by alcohol or metabolic syndrome is known to aggravate the course of specific liver diseases as in chronic hepatitis C [10]. Therefore, as e.g., in chronic hepatitis C in patients with metabolic risk factors or alcohol consumption, increased mortality is observed and is generally correlated with the development of liver cirrhosis and the occurrence of hepatocellular carcinoma (HCC, for an overview see [11]).

Despite the steadily increasing scientific interest in steatotic liver diseases, the molecular pathologic basis of such diseases has not been completely understood [12]. Generally, as in ASH and NASH, lipotoxic injury is believed to be the main mechanism that causes ballooned hepatocytes, inflammation and subsequent perisinusoidal fibrosis [13,14,15,16,17]. Ballooned hepatocytes, a sort of hepatocyte degeneration with swelling, enlargement, the breakdown of LDs and alterations of the cytokeratin network, present a hallmark of steatohepatitis irrespective of the cause [18]. Mediated by inflammatory factors coming from the ballooned cells, hepatic stellate cells (HSC, formerly designated as “fat-storing cells”, Ito cells, or perisinusoidal cells),which are a cell type known to store vitamin A in LDs in the space of Disse, are activated leading to the typical perisinusoidal fibrosis [19,20]. As a typical but not specific feature of ALD, Mallory–Denk bodies, cytoplasmic inclusions related to cytokeratins appear [21,22], which are nowadays regarded as a protective reaction of the respective hepatocytes. In contrast to ASH and NASH, in chronic viral hepatitis, besides hepatocyte apoptosis, parenchymal and portal inflammation is typical [3]; whereas, steatosis has long been regarded to be specific to HCV infections, especially in HCV genotype 3, where microvesicular, non-zonal-bound steatosis has been described. Yet, interestingly, aberrations in lipid metabolism have also been reported in the hepatocytes of chronic hepatitis B [23]. At the molecular level, HCV requires LDs for viral replication [24], and HCV core protein and non-structural protein (NS)5A have been shown to associate with LDs [25], for review see [26]. Despite modern viral therapies, patients that by the time of treatment have advanced liver disease are still at high risk of developing HCC, even if the virus has been eliminated [27]. Thus, liver cirrhosis and HCC in patients with chronic hepatitis C still pose immense health problems worldwide. In addition, the occurrence of other types of viral hepatitis, such as those related to COVID-19 [28,29] and especially the present accumulation of cases of adenoviral hepatitis in children [30,31] pose fresh challenges.

Steatosis is defined as the accumulation of LDs in hepatocytes that exceeds 5% of the liver parenchyma. LDs are cell organelles for the storage of triacylglycerides (TAGs) that are covered by a phospholipid monolayer in which amphiphilic proteins of the perilipin family are integrated [32]. On the cytoplasmic surface, LDs are attached to the intermediate filament network [33,34,35]. LDs are pivotal for milk secretion in the mammary gland, but LDs in other cell types such as hepatocytes and adipocytes also exhibit striking similarities in morphology and protein content [36]. In adipogenesis, LDs have been shown to be versatile and highly dynamic cell organelles [37,38,39]. Perilipins play a major role in the biogenesis, structure and degradation of LDs among different species (concerning Drosophila see e.g., [40]). In mammals, the perilipin family consists of five members: perilipin 1–5 (formerly designated as: perilipin [41], adipophilin (ARDP, ADFP) [42,43], TIP47 [44], S3-12 [45] and MLDP (OX-PAT) [46,47]). While perilipin 1 and 2 are constitutively localized on the LD surface (cPATs), perilipin 3–5 associate with LDs only under certain conditions (ePATs) [48]. Imbalances in LDs are implicated in various diseases [49], most often in liver diseases [50,51,52] as could also been shown functionally in mice (for perilipin 2 k.o.-mice see [53]). In addition, alterations in lipid metabolism in malignant tumours have been described that may parallel alterations in glucose metabolism known as the Warburg effect [54] and also LD accumulation has been described to be a frequent phenomenon of neoplastic steatogenesis [55,56]. In the last few years, we were able to show that all perilipins are found in the human liver and are differentially expressed with respect to normal and diseased steatotic liver, size of LDs, and localization in hepatocytes as well as in HSC [50], for an overview see [52]. Perilipin 2 is a well-suited marker for the detection of micro- or macrovesicular LD accumulation in the liver irrespective of cause ([49], for a parallel study on perilipin 2 in ballooned hepatocytes see [57]), whereas perilipin 1 marks chronic hepatic steatosis and is absent in acute steatosis [58].

So far, studies on LD accumulation in the human liver in situ focused mainly on NAFLD/ASH and ALD. In chronic hepatitis C, previous studies were able to demonstrate the importance of lipid droplet accumulation, especially for genotype 3 and of perilipin 2 for lipid droplets in HCV replication [59,60,61,62]; however, knowledge on steatosis pattern in chronic viral hepatitis is restricted, presumably due to a lack of representative patient material. The aim of the present study was to analyse LD-associated proteins of the perilipin family in a large collection of formalin-fixed, paraffin-embedded (FFPE) liver biopsies of viral hepatitis C of different grades and stages.

## 2. Results

### 2.1. LDs and Perilipin Expression in Liver

To validate the occurrence of LDs and associated proteins of the perilipin family in normal and in diseased livers, as shown in own previous studies [50], confocal laser scanning fluorescence microscopy of cryopreserved normal bovine and human livers and steatotic human livers was used. In normal livers, irrespective of the species analysed, LDs and associated proteins were found mainly in the non-parenchymal HSCs (for perilipins 1 and 2 in bovine liver see Figure 1a,b);whereas, in human NAFLD, LDs were detected mainly in hepatocytes, and less were detected in non-parenchymal cells/HSCs (Figure 1d–f), although fluorescence microscopy of the human liver was in part hindered by large amounts of lipofuscin found in the hepatocytes of older patients. As lipofuscin hindered the evaluation of normal human liver specimens in immunofluorescence microscopy, a collection of 30 normal human livers without or with only mild steatosis and without inflammation or fibrosis were stained immunohistochemically against perilipins 1 and 2 to validate the results obtained by immunofluorescence microscopy. Perilipin 1 was only present in steatotic hepatocytes as in NAFLD, but not in the normal human liver (for more comprehensive analyses compare previous studies [50,58]), whereas all other perilipins (perilipins 2–5) were present in normal as well as steatotic liver specimens. When LD stains such as Nile red were compared to perilipin staining, perilipins 1 and 2 constitutively covered the margin of Nile red-positive LDs in tissues as well as in cell culture, whereas perilipins 3–5 covered LDs only in certain conditions, such as after oleate treatment, as shown in the hepatocellular carcinoma cell line PLC/PRF/5/Alexander with known HBV integration (Figure 1c).

### 2.2. Steatosis in Chronic Hepatitis C

LDs play an important role in HCV replication and the steatosis of a HCV-infected liver is known to confer an unfavourable prognosis [12,24]. We have therefore evaluated the LD pattern in liver biopsies with chronic hepatitis C in comparison to clinical parameters in a large collection of 231 liver biopsies with known HCV genotype (Table A1 and Table A2). In order to determine the extent of steatosis associated with chronic hepatitis C, consecutive sections of liver biopsies were stained with H&E and PAS as well as immunohistochemically for perilipins (Table 1). In 60% of HCV-liver biopsies, relevant liver steatosis of >5% was detected (mostly mild steatosis) whereas in 19% of liver biopsies, a moderate to high degree of steatosis was observed (Table 2). A significant amount of microvesicular steatosis in hepatocytes was found with perilipin 1 and 2 immunostains that could not be detected in H&E and PAS staining. In liver biopsies without steatosis, perilipin 1 was negative, whereas perilipin 2 localized to the rim of LDs in non-parenchymal cells, most presumably HSCs, and only to a few LDs in the hepatocytes (Figure 2a). In liver biopsies with at least mild steatosis, perilipin 1 was expressed predominantly in zone 3. In liver biopsies with moderate to strong steatosis, perilipin 2 was only detected in hepatocytes, but not in non-parenchymal cells. In addition, in a subset of 25 liver biopsies, perilipins 3 and 5 were detected immunohistochemically but showed a predominantly diffuse cytoplasmic staining in hepatocytes, irrespective of the presence or absence of steatosis, and were therefore excluded from further analyses.

When the amount of steatosis was correlated with HCV genotypes, biopsies of patients with HCV genotype 3a showed a higher content of steatosis than those of patients with HCV genotype 1b (*p* = 0.014 according to the Wilcoxon test, Figure 2b, Table 1). Further analyses of HCV genotypes and steatosis showed a non-significant *p*-value of *p* = 0.453 in the Kruskal–Wallis test. In the multiple testing correction (FDR), the *p*-value between genotype 1b and 3a was *p*.adj = 0.95, although the Wilcoxon test was significant between these two genotypes (Figure 2b). The non-significant values in the Kruskal–Wallis test and in the multiple testing correction (FDR) could be explained by the high number of groups and consequently the small sample sizes. However, the degree of steatosis was not significantly correlated with the HCV viral load in the serum. As expected, both the degree of steatosis as determined by H&E and detected by perilipin 2 staining correlated significantly with the transaminase and bilirubin as well as y-GT levels in serum (Appendix A), but not with both alkaline phosphatase and choline esterase levels in serum (*p*-value > 0.05) (Appendix A). As expected, the grade of steatosis determined by H&E staining correlated significantly with the grade of steatosis determined by perilipin 2 immunohistochemistry (Appendix A).

In summary, in our study, by using immunohistochemistry staining for perilipins 1 and 2, we were able to confirm that chronic hepatitis C, particularly when caused by HCV genotype 3a, is associated with steatosis, as has been demonstrated before [10]. However, in our collection of samples, a higher degree of steatosis was not significantly correlated with higher overall viral load.

### 2.3. Steatotic Foci in Chronic Hepatitis C

Whereas perilipins 1 and 2 regularly localized to rim-like structures corresponding to the surface of different sized LDs, 36% of liver biopsies with chronic hepatitis C (82/227 cases) showed aberrant patchy steatosis with intense cytoplasmic and microvesicular LD staining in irregularly arranged cohesive clusters of hepatocytes, sharply demarcated from adjacent unstained hepatocytes (Figure 3). These microvesicular steatotic foci contained upwards of 20–30 hepatocytes with a mean of 11 hepatocytes and were more often found in liver biopsies with mild or moderate steatosis; whereas, in liver biopsies with strong steatosis, perilipin-stained LDs were more evenly distributed among hepatocytes without any zonal or patchy distribution (Figure 3a,b). In most cases, these perilipin 1- and 2-positive microvesicular steatotic foci were not detectable in H&E or PAS-stained consecutive serial sections (Appendix A). Microvesicular steatotic foci were better discernable in perilipin 2 than in perilipin 1 stains. When the presence of perilipin-positive microvesicular steatotic foci was correlated with the degree of steatosis as estimated by H&E or perilipin 2 immunostain, liver biopsies with perilipin-positive steatotic foci showed a significantly higher content of steatosis (11.67% and 8.06% in comparison to 6.85% and 6.14%; *p* < 0.05, Wilcoxon-text, Wilcoxon test effect sizes of 0.57 and 0.31; Figure 3c,d, Table 2).

Therefore, in chronic hepatitis C, small perilipin-positive microvesicular steatotic foci are present in more than 1/3 of biopsies correlating with overall steatosis in the liver parenchyma.

### 2.4. Steatotic Foci Are Areas of Focal Metabolic Dysregulation

To determine the biological significance of these perilipin-positive microvesicular steatotic foci, we investigated whether these steatotic foci may be a sign of viral infection. Therefore, in a subset of 72 liver biopsies with chronic hepatitis C, we performed immunohistochemistry with antibodies against HCV core antigen. A positive staining for the HCV core antigen could only be detected in a few samples with an acute hepatitis C infection with a high viral load, but not in the other liver biopsies in our collection (data not shown). In addition, using RNA CISH for HCV genotypes 1a and 1b, we could only demonstrate viral infection in a few liver biopsies of patients in some hepatocytes which were also strongly positive for perilipins 1 and 2, mostly with HCV signals at the border of LDs or diffusely in the cytoplasm (Appendix A, see inserts). However, we were not able to correlate HCV-infected cells with perilipin-positive microvesicular steatotic foci in the majority of our cases due to the low sensitivity of RNA CISH and the immunohistochemistry.

As the demonstration of viral infection in situ is known to be exceptionally difficult in chronic hepatitis C, we evaluated thirty-two liver biopsies and one liver resection specimen from patients with chronic hepatitis B with a known viral load (Appendix A) using immunohistochemistry with antibodies against HBs and HBc antigens. However, perilipin 1- and 2-positive steatotic foci only partially overlapped with areas of positive HBc- or HBs-staining (Figure 4a). In addition, control liver biopsies with autoimmune hepatitis (average age of patients = 13 years) demonstrated perilipin-positive steatotic foci in three out of of twelve (25%) cases (Figure 4b); whereas, in control normal livers with no or only mild steatosis from patients with liver metastases with no known viral or other liver diseases, perilipin-positive foci could be detected in 10 out of 29 cases (34%), maybe due to the larger area analysed when compared to liver biopsies, in which naturally only small areas were analysed. Thereby, microvesicular steatotic foci were also found in apparently normal livers and were not confined to chronic hepatitis C or the other cases of chronic liver diseases analysed in the control.

In order to further characterize perilipin-positive microvesicular steatotic foci, we performed electron microscopy, histochemistry and immunohistochemistry for marker proteins of cell proliferation and early tumorigenesis in consecutive liver sections of the same biopsies. In most cases, intensively stained perilipin 2-positive foci were also at least partly visible in perilipin 1 stainings, whereas only in a few cases of non-viral liver disease, with PAS and H&E stainings, a faint discoloration was found (Figure 4c), but no PAS-positive glycogenosis as observed in clear cell foci described by Ribback and coauthors [63] was detected. Using transmission electron microscopy of the same perilipin-positive steatotic foci from FFPE liver tissue, hepatocytes filled with many minute to small LDs were present with a sharp demarcation of hepatocytes without any LDs (Figure 4d). In one case, mitochondrial aberrations with crystal formations were found in the perilipin-positive microvesicular steatotic focus as well as in the surrounding liver parenchyma, which may reflect hepatocyte injury most probably due to previous chemotherapy in this specific patient (Appendix A). No other significant ultrastructural aberrations could be noted; there were no signs of ballooning and no condensation of the filament network. Ki67-proliferation rate was generally below 1% and not significantly elevated when compared to the surrounding liver parenchyma, and no aberrant expression of p53, glypican 3, HSP70, phospho-mTor, Akt, β-catenin, E- or N-cadherin was detected (data not shown); markers were reported to be deregulated in (early) tumorigenic lesions. With respect to zonation as tested by glutamine synthetase, perilipin-positive steatotic foci were not strictly confined to certain hepatic zones but appeared to be localized randomly. In one case, glutamine synthetase showed negativity in a perilipin-positive steatotic focus that was situated in zone 2 to 3 (Figure 4c). In nine liver resections showing clear cell foci, these foci did not show aberrant perilipin staining and perilipin-positive microvesicular steatotic foci did not show an overlap with clear cell foci as tested by PAS stain (Appendix A).

In summary, our results suggest that perilipin-positive microvesicular steatotic foci are especially frequent in chronic hepatitis C; however, patchy steatosis is also detected at a lower rate in other chronic liver diseases.

## 3. Discussion

This is the first study to comprehensively evaluate steatosis as determined by conventional microscopy as well as immunohistochemical perilipin stainings in chronic hepatitis C in a large clinicopathologic study of liver biopsies and liver resection specimens of 306 patients overall, including 231 liver biopsies of HCV patients as well as control livers with HBV, AIH and normal livers. As demonstrated in previous studies [49], perilipin 2 proved to be a particularly suitable marker for the identification of microvesicular steatosis not visible by conventional light microscopy. In this study, we could demonstrate differences in steatosis based on HCV genotypes in situ similar to those shown clinically and in vitro by other authors [64,65]. Apart from the genotype, steatosis did not correlate with the overall viral load, indicating that additional factors are responsible for LD accumulation [64]. Metabolic syndrome or alcohol abuse in patients with chronic hepatitis C is known to be a negative prognostic factor. Despite the use of novel direct antiviral agents resulting in HCV elimination in over 95% of cases, some patients with chronic hepatitis C develop NASH. Therefore, additional factors of lipotoxicity may be present in patients with progressive chronic hepatitis C [66,67]. As LDs are necessary for HCV replication [24,68], we speculate that lipotoxic injury may support progressive liver disease in chronic hepatitis C that may persist after antiviral treatment. In liver biopsies with no or only mild steatosis due to HCV or NAFLD, perilipin 2 staining was only detected in non-parenchymal cells, most presumably in HSCs, as already shown before [50]; however, in livers with moderate to strong steatosis, perilipin 2 was mainly detected in hepatocytes whereas perilipin 2-staining in non-parenchymal cells of the liver was diminished or absent, pointing to a pathogenic role of HSCs in the progression of chronic hepatitis C. HCV core protein has been shown to interact with ATGL and ABHD5 at perilipin 2-positive LDs in hepatocytes [69]. Perilipin 5 has been shown to alleviate HCV NS5A-induced lipotoxic injury in mice [70]. In our HCV collection, no statistically relevant differences in perilipin 5 staining were observed nor could we detect HCV core protein in chronic hepatitis with a low viral load. The coincidental discovery of chronic HCV and steatosis and especially the persistence of steatosis after HCV treatment argues for additional nutritional and behavioural factors in the respective patients (see e.g., [71,72]) that may need additional treatment (see e.g., [73]) to improve outcomes.

In addition, for the first time, perilipin 2-positive microvesicular steatotic foci of different sizes were detected in liver biopsies of chronic hepatitis C. From our data, we could not substantiate the finding that microvesicular steatotic foci may originate from a present or prior viral replication; however, LD accumulation as triggered by HCV may be the initial event in the formation of microvesicular steatotic foci [24]. In this study, only FFPE liver biopsies of patients with HCV were used, as no cryopreserved liver tissue of HCV-positive, non-cirrhotic patients was available due to biobanking regulations limiting the storage of cryopreserved infectious material. Perilipin-positive microvesicular steatotic foci were especially frequent in liver biopsies of patients infected with HCV but were not restricted to HCV-positive livers and were also found in liver biopsies of control patients with HBV, AIH and (apparently) normal livers. Steatotic foci seemed to appear less in normal young livers, but generally in livers with at least a mild to moderate steatosis, although this phenomenon may need to be further substantiated using a higher number of cases that includes more normal livers of young patients, which were naturally difficult to acquire from routine diagnostic material. We conclude that microvesicular steatotic foci may therefore represent areas of focal metabolic dysregulation triggered by liver injury such as HCV infection. These microvesicular steatotic foci may give rise to larger steatotic foci or areas of steatosis first described in 1980 under the term focal fatty change [74]. Larger areas of focal fatty change have been described using radiological means and may pose a differential diagnosis to tumorous or infectious liver lesions (see also [75,76,77]).

Perilipin-positive steatotic foci showed no significantly elevated proliferation rates, but one case demonstrated an altered zonal pattern of the wnt target glutamine synthetase. Altered metabolism with accumulations of lipids and/or glycogen have previously been shown in preneoplastic and early neoplastic foci in mice giving rise to HCC [78]. Ribback and coauthors were able to identify glycogenotic clear cell foci in human tissue that resembled foci in rat hepatocarcinogenesis and they presented evidence that glycogenotic foci are in fact very early preneoplastic lesions [63,79]. Cano and coauthors detected higher numbers of clear cell foci and viral infection in patients with non-cirrhotic, non-fibrolamellar HCC in young patients [80]. As well as glycogenated foci, steatotic foci have also been described in mice and are thought to demonstrate a significant overlap, therefore possibly representing two sides of the same coin [78]. Early HCC has been shown to frequently demonstrate fatty change [81,82]. In addition, our group could demonstrate increased perilipin 2 amounts in a large variety of malignant tumors when compared to the respective normal tissues [83] reminiscent of a Warburg effect [54]. In HCCs, increased amounts of microvesicular perilipin 2-positive LDs correlated positively with the Ki67-proliferation rate [83]. Both fatty acid and glycogen accumulation have been shown to be signs of an altered metabolism in non-neoplastic and neoplastic hepatocytes. Therefore, we cannot exclude the possibility that clear cell or steatotic foci may show metabolic dysregulation as a sign of regeneratory or even early preneoplastic alterations. In our view, clear cell or steatotic foci may represent a sign of liver injury providing susceptibility for liver tumorigenesis rather than being a true preneoplastic lesion. When taking into account the high frequency of clear cell and steatotic foci, even when assuming that clear cell or steatotic foci may be potentially preneoplastic lesions, the progression risk of these foci is expected to be extremely low. Further analyses are needed to determine the impact and importance of these foci.

Finally, our description of frequent, and so far, undescribed microvesicular steatotic foci in viral hepatitis was astonishing, especially when one takes into account the longstanding research on viral hepatitis, and we can conclude that there are still findings that have so far escaped notice. Meanwhile, the topic of viral hepatitis has currently become highly relevant due to the COVID-19-pandemic. Intriguingly, SARS-CoV-2 was rarely shown to lead to hepatitis in children [28], see also [29]. Since then, so far 1000 cases of a fulminant acute hepatitis in children were reported which in part necessitated liver transplantation [84]. These cases were presumed to have been caused by a combined adenovirus infection (human adenovirus 41; see [30,85]) and co-infection with adeno-associated virus 2 (AAV2, [86] Preprint; [31] Preprint), possibly in patients with a background of immunodeficiency or retarded immunological training due to hygienic measures during the COVID-19 pandemic. Future studies will therefore need to address different types of viral hepatitis, preferably using fresh or cryopreserved patient material to further delineate the reason for lipid perturbations frequently observed in viral hepatitis and the clinicopathologic impact of focal microvesicular steatotic lesions in viral hepatitis and other chronic liver diseases.

## 4. Material & Methods

### 4.1. Tissue and Collectives

Formalin-fixed, paraffin-embedded (FFPE) liver biopsies of 231 patients with chronic hepatitis C (Table A1 and Table A2/see Appendix B) were used. A total of 30 out of the 231 patients had an additional HBV infection or a history of HAV infection, 3 patients were coinfected with HIV. A total of 71 patients had a history of intravenous drug consumption. In addition, 32 liver biopsies and 1 liver operation specimen with chronic hepatitis B (Appendix A), and 12 liver biopsies with autoimmune hepatitis were used. As controls, the liver tissues of 30 patients with mild or no steatosis who underwent surgery for liver metastases or echinococcus granulosus infection were analysed. As additional control, snap-frozen normal bovine and human liver and steatotic human liver tissue from operation specimens taken from a liver metastasis were used. However, cryopreserved liver tissue of chronic viral hepatitis C was not available.

Tissue samples were provided by the biobank of the University Medical Center Mainz in accordance with the regulations of the tissue biobank and the ethics vote of Rhineland-Palatinate to B.K.S. (2020-15428, “Lipidtropfen und -assoziierte Proteine bei viralen Hepatitiden”) as well as the regulations of the ethics committee of the University of Heidelberg (206/2005 and 207/2005).

### 4.2. Immunohistochemistry and Histochemistry

Immunohistochemistry of FFPE liver biopsies as well as H&E and PAS stains were performed as previously described [50,83]. The antibodies used for immunohistochemistry are listed in Table 3.

### 4.3. Immunofluorescence Microscopy

Liver cryosections of approx. 5 μm thickness were air-dried for 1 h and fixed in 4% paraformaldehyde at room temperature for 20 min. Sections were then incubated with the primary antibodies diluted in PBS for 30 min and were then washed twice with PBS for 5 to 10 min each. Incubation with secondary antibodies took place for 30 min with two subsequent washes in PBS for 5–10 min. The secondary antibodies used were Alexa 488-coupled anti-mouse, -rabbit and -guinea pig IgG antibodies (MoBiTec, Göttingen, Germany) as well as the respective Cy3 and Cy5 coupled anti-mouse, -rabbit and -guinea pig IgG antibodies (Dianova, Hamburg, Germany). For the staining of LDs, Nile red (Sigma-Aldrich, St. Louis, MO, USA) was solubilized in ethanol at 1 mg/mL and then diluted from 1:500 to 1:5000 in PBS for the immunofluorescence staining of LDs. Sections were then rinsed in distilled water and mounted directly in Fluoromount G (Biozol Diagnostica, Eching, Germany). If staining for lipids was not necessary, tissue cryosections or cultured cells were fixed in methanol and acetone. Cultured cells of the line PLC/PRF/5/Alexander (ATCC CRL 8024) grown on glass coverslips to about 70% confluency were treated with 250 µM BSA-coupled oleate (Sigma Aldrich, St. Louis, MO, USA), washed repeatedly in PBS containing 2 mM MgCl2 at 37 °C and fixed with 2% paraformaldehyde in PBS for 10 min. Fixed cells were washed repeatedly with PBS for 30 min, and permeabilized in PBS with 0.05% Tween 20 for 10 min. Epifluorescence was performed using a Zeiss Axiophot photomicroscope (Zeiss, Jena, Germany). Confocal laser-scanning immunofluorescence microscopy was performed using a Zeiss LSM 510 microscope.

### 4.4. RNA Chromogen In Situ Hybridization (CISH)

RNA-CISH with specific probes directed against HCV genotype 1a and 1b was performed according to the recommendations of the manufacturer (Affymetrix, Santa Clara, CA, USA; QuantiGene ViewRNA ISH Cell Assay) [87].

### 4.5. Electron Microscopy

Transmission electron microscopy of formalin-fixed, paraffin-embedded (FFPE) liver tissue was performed in accordance with standard procedures. Images were taken with a 4k camera using a Jeol Jem 1400HC microscope (Akishima, Tokyo, Japan).

### 4.6. Evaluation of Immunohistochemistry

Immunohistochemical stainings were evaluated manually by S.S and B.K.S. Staining intensity was scored in accordance with the IRS score (product of intensity: 0 = negative, 1 = weak, 2 = moderate, 3 = strong staining, and percentage of positive cells: 1–9%: 1, 10–49%: 2, 50–79%: 3, 80–100%: 4, i.e., 0–12) [88]. In addition, steatosis was graded according to Kleiner [89]. Perilipin-positive steatotic foci were classified as negative for foci (category 0), positive for foci (category 1), or indefinite for foci (category 2). The grade and stage of chronic viral hepatitis was scored according to Desmet [90].

## Figures and Tables

**Figure 1 ijms-23-15456-f001:**
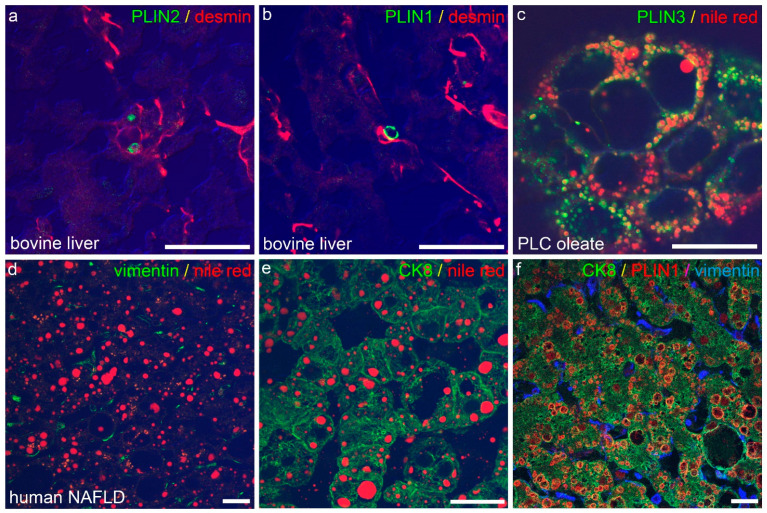
Immunofluorescence microscopy of LDs and perilipins in the liver. (**a**,**b**) Confocal laser scanning fluorescence microscopy of perilipin 2 ((**a**), PLIN 2, green) and perilipin 1 ((**b**), PLIN1, green) mark LDs in desmin (red)-positive, non-parenchymal HSC in cryopreserved normal bovine liver. Note absence of perilipin 1 and 2-positive LDs in parenchymal cells of normal bovine liver. (**c**) Perilipin 3 (PLIN3, green) localizes to the rim of Nile red (red)-positive LDs in oleate-treated cultured cells of the HCC-derived line PLC. (**d**–**f**) Accumulation of Nile red (red)-positive LDs in human NAFLD is attributed to fatty change in hepatocytes ((**e**), CK8, green), whereas HSC ((**d**), vimentin, green) show no Nile red-positive LDs. As an example, perilipin 1 (PLIN1, red) localizes to LDs in CK8 (green)-positive hepatocytes, but not to vimentin (blue)-positive HSC. Merge images in (**a**,**b**,**d**,**e**) include differential interfering contrast. Scale cars: each 25 µm.

**Figure 2 ijms-23-15456-f002:**
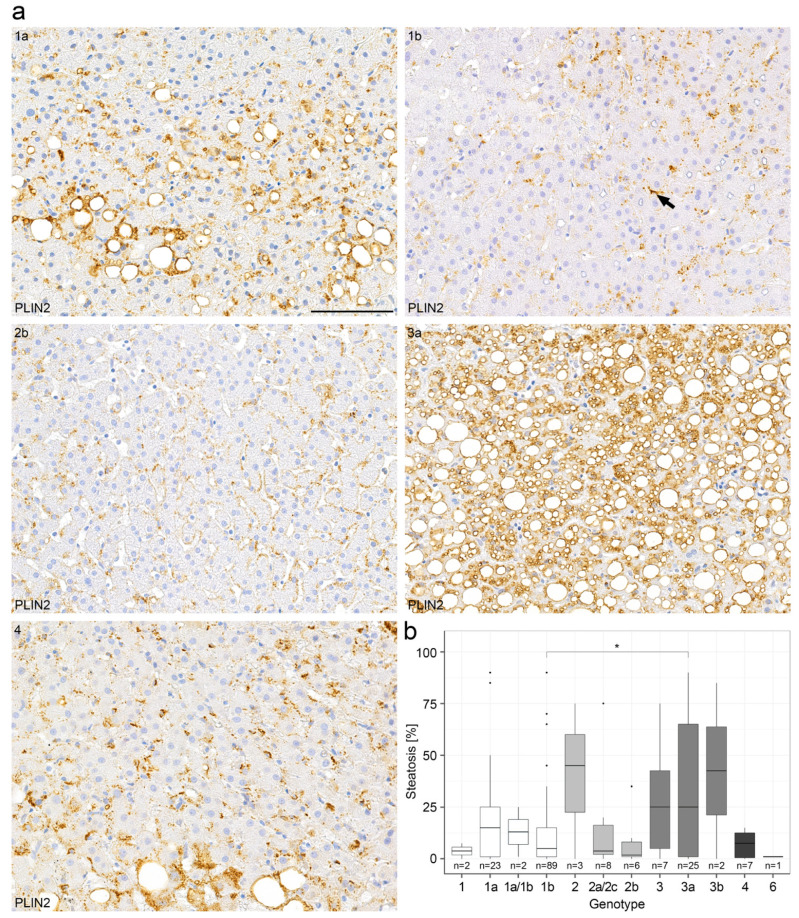
Perilipin 2 immunohistochemistry demonstrates different degrees of steatosis in chronic hepatitis C in relation to HCV genotype. (**a**) Representative immunohistochemical perilipin 2 (PLIN2) stains from a collection of 231 liver biopsies of patients with chronic hepatitis C with respect to the different HCV genotypes 1a,1b,2b,3a and 4. The arrow depicts perilipin-2 positivity in an HSC. (**b**) Correlation of the degree of steatosis as determined by conventional H&E stain with respect to the HCV genotypes (boxplot). Wilcoxon test with *p* = 0.014. *p*-values < 0.05 (*) were considered significant. Scale bar = 100 µm. *n* = numbers of cases evaluated per genotype.

**Figure 3 ijms-23-15456-f003:**
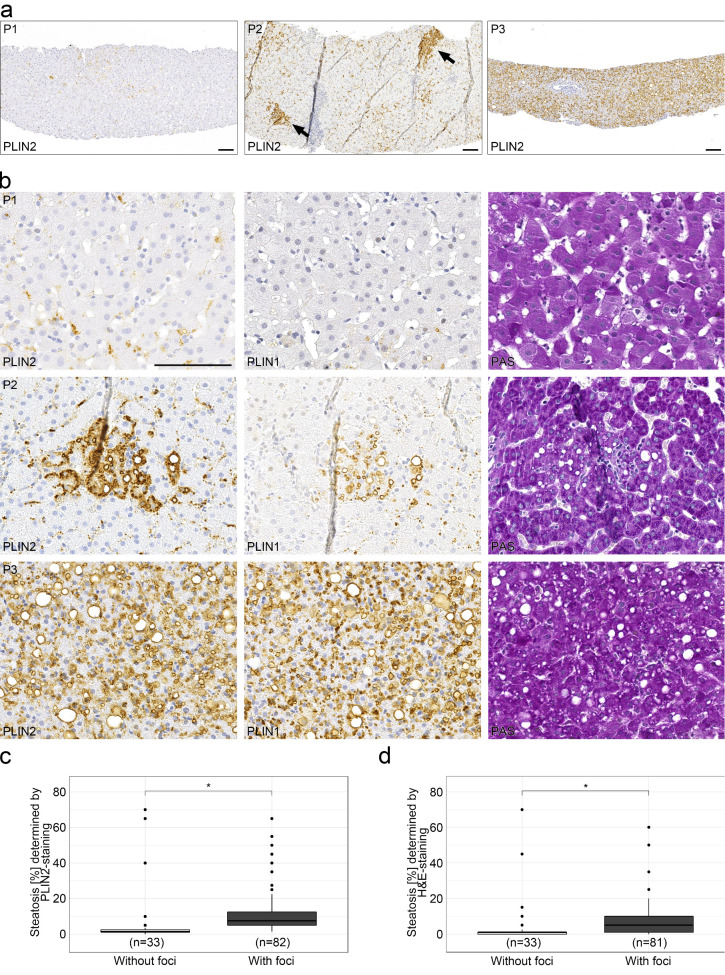
Appearance of perilipin-positive microvesicular steatotic foci with chronic hepatitis C infections. (**a**) Overview of representative immunohistochemical stains for perilipin 2 (PLIN2) in liver biopsies from 3 different HCV patients with no (P1), minor (P2) and a moderate (P3) degree of steatosis. Arrows indicate perilipin 2-positive steatotic foci. Scale bar: 100 µm. (**b**) Higher magnification of the liver biopsies of the same patients with respective immunohistochemical stains for perilipins 1 and 2 as well as PAS staining. Scale bar = 100 µm. (**c**,**d**) Occurrence of steatotic foci with respect to the degree of steatosis (boxplot). Foci were identified by perilipin 2 (**c**) or H&E staining (**d**). Wilcoxon test with *p* < 0.05 (*) were considered significant.

**Figure 4 ijms-23-15456-f004:**
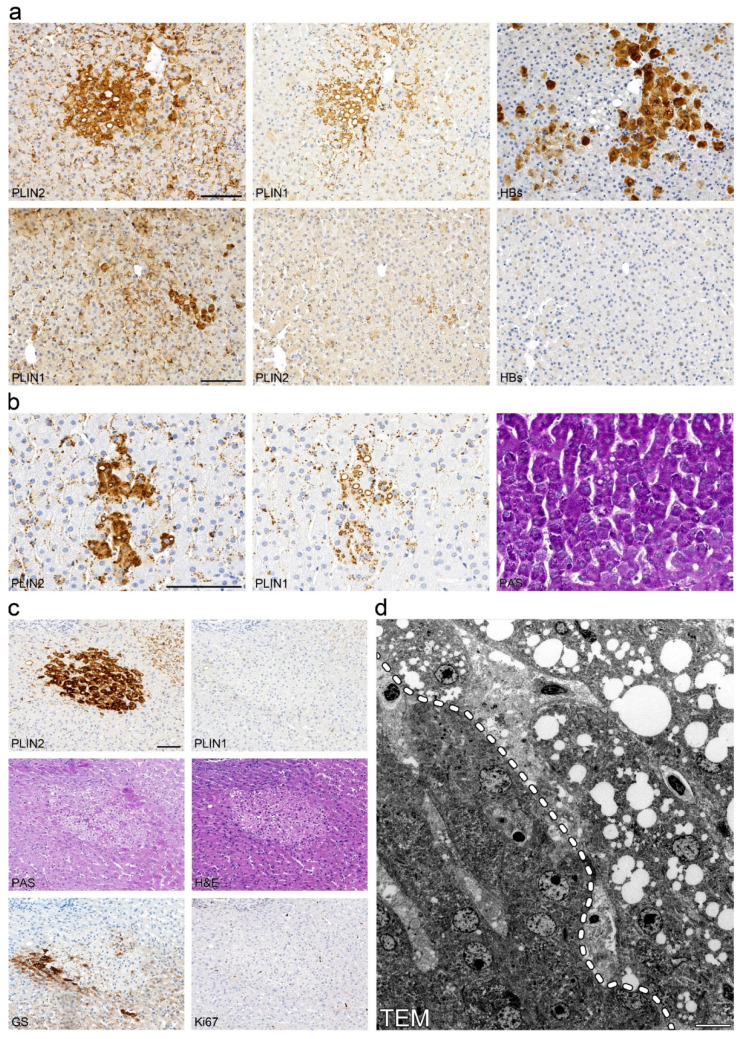
Perilipin-positive microvesicular steatotic foci do not overlap with virus-infected clusters and may also be found in other chronic liver diseases. (**a**) Immunohistochemical analysis of perilipins 1 and 2, and hepatitis B surface antigen (HBs) in consecutive sections of liver biopsies with chronic hepatitis B. Steatotic foci with (upper panel) or without (lower panel) detectable HBs-Ag positive clusters are shown. (**b**) Immunohistochemistry for perilipins 1 and 2 as well as PAS stain of consecutive liver sections from a patient with autoimmune hepatitis. (**c**) Immunohistochemistry for perilipin 1, perilipin 2, glutamine synthetase and Ki67 as well as PAS and H&E stain of consecutive sections from a liver operation specimen from a patient with liver metastasis with a neuroendocrine tumor. (**d**) Transmission electron microscopy of steatotic focus shown in (**c**). The dashed line (left panel) indicates the boundary of the focus. Scale bar = 10 µm.

**Table 1 ijms-23-15456-t001:** Perilipins in different HCV genotypes.

Parameter	HCV Genotypes					
	1	2	3	4	6	Not Tested/Unknown
PLIN2 score [0–3]					-	
mean (SD)	1.85 (0.79)	1.75 (0.73)	2.27 (0.79)	1.79 (0.49)		1.92 (0.74)
*N*	114	16	34	7		56
PLIN2 foci share [%]	35.09	50	23.53	57.14	-	39.29
*N*	114	16	34	7		56
PLIN1 score [0–3]					-	
mean (SD)	1.15 (0.5)	1.3 (0.56)	1.66 (0.71)	0.71(0.27)		1.26 (0.53)
*N*	114	16	34	7		56
PLIN1 foci share [%]	29.82	18.75	11.76	0	-	16.07
*N*	114	16	34	7		56
Foci share of all hepatocytes [%]					-	
mean (SD)	13 (15)	8 (4)	45 (38)	10 (6)		23 (26)
*N*	51	10	16	4		33
Steatosis in PLIN2 [% of parenchyme area]						
mean	15.72	16.15	35.71	14.36	1	23.24
(SD)	(20.94)	(24.07)	(32.63)	(15.59)	-	(24.47)
*N*	115	17	34	7	1	55
Steatosis in H&E [% of parenchyme area]						
mean	12.54	17.09	34.03	6.93	1	18.15
(SD)	(18.46)	(25.38)	(32.69)	(6.72)	-	(22.95)
*N*	116	17	34	7	1	55

**Abbreviations:** HCV: hepatitis C virus; PLIN1: perilipin 1; PLIN2: perilipin 2; SD: standard deviation; *N*: number. **Further explanations:** PLIN1/2 foci share: percentage of liver biopsies with detectable foci; Foci share of all hepatocytes: foci proportion compared to all hepatocytes within a biopsy.

**Table 2 ijms-23-15456-t002:** Grade of steatosis and percentage of steatotic foci in chronic hepatitis C.

Grade of Steatosis in % (as Evaluated in H&E)	Number of Cases	Percentage of PLIN2-Positive Foci [%]	Percentage of PLIN1-Positive Foci [%]
<5	89	41.57	25.84
5–33	94	42.55	27.66
33–66	27	14.81	3.70
>66	16	0	0

**Abbreviations:** PLIN1: perilipin 1; PLIN2: perilipin 2.

**Table 3 ijms-23-15456-t003:** Antibodies used for immunohistochemistry and immunofluorescence microscopy.

Antibody (Clone)	Immunoglobulin Class	Provider
Akt (C67E7)	Monoclonal rabbit	Cell Signaling Technology, Danvers, MA, USA
Beta-Catenin (β-Catenin-1)	Monoclonal mouse	Agilent, Santa Clara, CA, USA
CK8/18 (GP11)	Polyclonal guinea pig	PROGEN, Heidelberg, Germany
E-cadherin (36/E-Cadherin)	Monoclonal mouse	BD Biosciences, San Jose, CA, USA
Desmin (D9)	Monoclonal mouse	PROGEN
Desmin	Polyclonal rabbit	PROGEN
L-FABP1	Polyclonal rabbit	Sigma-Aldrich, St Louis, MO, USA
Glutaminsynthetase (GS-6)	Monoclonal mouse	Cell Marque/Sigma Aldrich
Glypican 3 (GC33)	Monoclonal mouse	Agilent
HCV Core antigen (C7-50, sc-57800)	Monoclonal mouse	Santa Cruz Biotechnology, Dallas, TX, USA
Heat shock protein 70 (W27)	Monoclonal mouse	Santa Cruz Biotechnology
Hepatitis B surface antigen (AI0FI)	Monoclonal mouse	DCS Innovative Diagnostics, Hamburg, Germany
Ki-67 (MIB-1)	Monoclonal mouse	Agilent
N-cadherin (36/N-Cadherin)	Monoclonal mouse	BD Biosciences
p53 (DO-7)	Monoclonal mouse	Agilent
Perilipin 1 (PERI 112.17)	Monoclonal mouse	PROGEN
Perilipin 1 (GP29)	Polyclonal guinea pig	PROGEN
Perilipin 1A	Polyclonal rabbit	Sigma Aldrich
Perilipin 2 (AP 125)	Monoclonal mouse	PROGEN
Perilipin 2 (GP40, GP41)	Polyclonal guinea pig	PROGEN
Perilipin 3 (GP30, GP32)	Polyclonal guinea pig	PROGEN
Perilipin 4 (GP38)	Polyclonal guinea pig	PROGEN
Perilipin 5 (GP31)	Polyclonal guinea pig	PROGEN
Phospho-mTor (49F9)	Monoclonal rabbit	Cell Signaling Technology
Vimentin (3B4, V9)	Monoclonal mouse	PROGEN

## Data Availability

Data may be made available on request.

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
