# Peer review of "Lipid Droplet-Associated Proteins Perilipin 1 and 2: Molecular Markers of Steatosis and Microvesicular Steatotic Foci in Chronic Hepatitis C"

_ijms, 2022, doi:10.3390/ijms232415456_

Round 1
Reviewer 1 Report
The study of Schelbert and colleagues investigated the presence and distribution of lipid droplet (LD)-associated perilipin 1 (PLIN1) and 2 (PLIN2) proteins in liver biopsies from patients chronically infected with various genotypes of hepatitis C virus (HCV), who developed different grades of steatosis (lipid droplet accumulation in the liver).
A strong point of this study is that it relies on liver biopsies from a large cohort of patients with chronic hepatitis C (231 samples) that have been analyzed to score several parameters: steatosis grade by hematoxilin & eosin (H&E) staining and/or periodic acid-schiff (PAS) staining, percentages of PLIN1- and PLIN2-positive foci, HCV RNA by in situ hybridization.
However, the manuscript suffers from substantial lack of clarity, in the description of both the aims of the study and the data. The Abstract does not faithfully reflect data shown and go beyond them. Furthermore, appropriate controls are missing in several experiments, and conclusions are not always supported by the data shown. Altogether, the concluding messages delivered are confusing or misleading, including the paper's title.
Below are some precise, yet non-exhaustive points that support the reviewer's evaluation.
- The Introduction is not focused. Authors list unrelated features targeting chronic hepatitis B, chronic hepatitis C, steatosis, hepatocellular carcinoma, viral attributes in a succession of sentences (e.g. page 2, lines 76-85) without clearly underlying their point. The first sentence of the Introduction (page 2, lines 51-52) "Steatosis [....] plays an important role in steatotic liver diseases [...]" should be clarified.
- In the Introduction, it is stated that no previous study focused on lipid droplet accumulation in the liver of HCV chronically infected patients in relation with pathological steatosis (page 3, lines 115-117), whereas some prior publications have dealt with this issue and deserve to be cited (e.g. Campana et al., J. Viral Hep. 2017; Zubair et al., Saudi J Gastroenterol. 2009; Piodi et al., Hepatology 2008; Boulant et al., Traffic, 2008).
- The aims of the study are set out as "[...] to analyze LD-associated proteins in a large collective of [...] liver biopsies of viral hepatitis C [...] in comparison to other types of viral hepatitis" (page 3, lines 117-120). There are actually no quantitative data shown in the paper with respect to "other viral hepatitis", nor statistical comparisons of cohorts. However, it is puzzling that such quantitative values are summarized in the Abstract. Therefore, data presented do not match the stated aims. This discrepancy should be solved.
- Expression of PLIN 1 and PLIN 2 was comparatively sought in normal liver and steatotic liver in Fig. 1. However, while the manuscript's Abstract states that "Perilipin 1 and 2-positive microvesicular steatotic foci were observed in 36% of HCV liver biopsies [...], but not at all in normal livers of young patients", Fig. 1 provides controls of bovine liver and not human liver. In the text, prior publications of the authors are cited to refer to data in normal human liver. At the minimum, authors should explain why there were unable to provide normal human liver control alongside HCV infected livers in the present study and correct the statement in the Abstract which is not derived from data of this study.
- It is stated that immunohistochemical perilipin staining is indicative of microvesicular steatosis not visible by conventional light microscopy following H&E / PAS staining (page 10, lines 295-297; page 1, lines 158-160). However, there is no correlation graph shown in support of this statement with the biopsies presented in this study.
- Content of Table 1 is not described in a comprehensible way. What do "PLIN1/PLIN2 cluster share (%)" and "Cluster share of all hepatocytes (%)" refer to? There are huge SD values and no statistical analysis is provided in support of concluding to higher content of steatosis in biopsies of patients with HCV genotype 3.
- Fig. 2b: By H&E staining, percentage of steatosis is higher with HCV genotype 3 chronic infection than with the majority of non-3 genotypes, which is in agreement with previously published information. Whether 1a/1b and 2a/2c are HCV recombinants or pooled subtypes is unclear. In this respect, the high steatosis rate for genotype 2 versus 2a/2c and 2b is puzzling. The number of biopsies examined in each genotype group should be indicated. A correlation graph of PLIN2 immunostaining (Fig. 2a) versus H&E staining (Fig. 2b) would objectify the conclusions.
- The percentages of PLIN1- and PLIN2-positive foci seem inversely correlated to the severity of steatosis (Table 2), which appears discrepant with images shown in Fig. 3b. Could the authors comment on this?
- The reviewer appreciates the difficulty to perform in situ labelling of HCV proteins or RNA in chronically infected livers. However, some conclusions are provided on the basis of HCV RNA in situ hybridization (page 7, lines 226-230), whereas no RNA label is detectable in Suppl. Fig. 3. Could the authors add arrows to refer to what they identify as viral RNA in this figure?
- Mitochondrial alterations in Fig. 4d should be pointed.
- Reference to Fig. 4d for the liver zonation issue (page 10, line 283) does not seem appropriate. There is no illustration to support these conclusions.
- What do the lines refer to in the non parametric Spearman correlation graphs (Suppl. Figs. 1 & 2)?
In conclusion, this manuscript requires profound remodeling and additional data analyses to convey clear aims and convincing conclusions with respect to the interesting question it tackles.
Author Response
Point-to-point response to Reviewer 1:
The study of Schelbert and colleagues investigated the presence and distribution of lipid droplet (LD)-associated perilipin 1 (PLIN1) and 2 (PLIN2) proteins in liver biopsies from patients chronically infected with various genotypes of hepatitis C virus (HCV), who developed different grades of steatosis (lipid droplet accumulation in the liver).
A strong point of this study is that it relies on liver biopsies from a large cohort of patients with chronic hepatitis C (231 samples) that have been analyzed to score several parameters: steatosis grade by hematoxilin & eosin (H&E) staining and/or periodic acid-schiff (PAS) staining, percentages of PLIN1- and PLIN2-positive foci, HCV RNA by in situ hybridization.
However, the manuscript suffers from substantial lack of clarity, in the description of both the aims of the study and the data. The Abstract does not faithfully reflect data shown and go beyond them. Furthermore, appropriate controls are missing in several experiments, and conclusions are not always supported by the data shown. Altogether, the concluding messages delivered are confusing or misleading, including the paper's title.
We have modified the title of the manuscript accordingly and rewritten large parts of the manuscript for better clarification. The controls were in fact appropriate, yet, we had to recognize, that liver specimens designated to be “normal” with respect to steatosis, fibrosis and inflammation, yet from patients with for example colorectal liver metastasis, showed microvesicular steatotic foci as well, and only young livers had less microvesicular steatotic foci. With respect to HCV, in addition, also other causes of viral hepatitis or non-viral hepatitis were used in control. We have explained the study design with this respect in more detail to make the point clear.
Below are some precise, yet non-exhaustive points that support the reviewer's evaluation.
- The Introduction is not focused. Authors list unrelated features targeting chronic hepatitis B, chronic hepatitis C, steatosis, hepatocellular carcinoma, viral attributes in a succession of sentences (e.g. page 2, lines 76-85) without clearly underlying their point. The first sentence of the Introduction (page 2, lines 51-52) "Steatosis [....] plays an important role in steatotic liver diseases [...]" should be clarified
The reviewer is right, the first sentence has been clarified. In addition, we have rewritten the introduction for better clarity.
- In the Introduction, it is stated that no previous study focused on lipid droplet accumulation in the liver of HCV chronically infected patients in relation with pathological steatosis (page 3, lines 115-117), whereas some prior publications have dealt with this issue and deserve to be cited (e.g. Campana et al., J. Viral Hep. 2017; Zubair et al., Saudi J Gastroenterol. 2009; Piodi et al., Hepatology 2008; Boulant et al., Traffic, 2008).
We thank the reviewer for his comments and have added the references accordingly.
- The aims of the study are set out as "[...] to analyze LD-associated proteins in a large collective of [...] liver biopsies of viral hepatitis C [...] in comparison to other types of viral hepatitis" (page 3, lines 117-120). There are actually no quantitative data shown in the paper with respect to "other viral hepatitis", nor statistical comparisons of cohorts. However, it is puzzling that such quantitative values are summarized in the Abstract. Therefore, data presented do not match the stated aims. This discrepancy should be solved.
We thank the reviewer for his comments and have improved this point. In fact, the numbers of other cases of other (acute and chronic) viral and non-viral hepatitis cases are less, namely of cases with HBV (33), normal livers (30) as well as AIH (12) as the focus was on HCV (with 231 liver biopsies). Therefore, no statistical correlations have been performed with this respect. Yet, the qualitative comparison of HCV to other causes of viral or non-viral hepatitis was important, especially for the incidence of microvesicular steatotic foci. Although especially frequent in chronic hepatitis C, also cases of HBC or AIH, and apparently normal livers of aged patients displayed microvesicular steatotic foci, so we needed to conclude that microvesicular steatotic foci are not specific to HCV.
- Expression of PLIN 1 and PLIN 2 was comparatively sought in normal liver and steatotic liver in Fig. 1. However, while the manuscript's Abstract states that "Perilipin 1 and 2-positive microvesicular steatotic foci were observed in 36% of HCV liver biopsies [...], but not at all in normal livers of young patients", Fig. 1 provides controls of bovine liver and not human liver. In the text, prior publications of the authors are cited to refer to data in normal human liver. At the minimum, authors should explain why there were unable to provide normal human liver control alongside HCV infected livers in the present study and correct the statement in the Abstract which is not derived from data of this study.
There seem to be some misunderstanding, which we would like to resolve. In fact, quantitatively large amounts of normal control livers were used in this study (n=30), in means of apparently normal liver specimens from e.g. operation specimens of patients with colorectal liver metastases with respect to absence of inflammation, no or only mild steatosis, and absence of fibrosis. Yet, also in about 1/3rd of these normal livers which were analysed as whole slides, especially in aged patients, perilipin 1 and 2 stained microvesicular steatotic foci in respective operation specimens. The relatively high amounts of foci in normal livers may be due to the larger area analysed in operation specimens when compared to liver biopsies analysed in HCV. In younger patients or in children with autoimmune hepatitis, less foci were stained. Of course, the numbers of perfectly normal human young livers were in fact scarce due to ethic restrictions, our and also other biobanks do not harbour normal human livers from children, adolescents or young adults. The reason is, that for diagnostic purposes, no operation specimens of biopsies of normal livers are taken due to the associated mortality of liver surgery / biopsy.
Immunofluorescence microscopy of perilipin 1 and 2 was done in human liver and livers of other species with respect to localization to hepatic stellate cells and/or hepatocytes. In figure 1, human liver is not shown due to the high content of lipofuscin making interpretation of fluorescence stainings difficult. Yet, the results are similar to those observed in other species. No data of previous studies were used, yet for more comprehensive studies concerning immunoblot, RNA analysis, colocalization studies, staining of perilipins at lipid droplets, we refer to previous studies.
- It is stated that immunohistochemical perilipin staining is indicative of microvesicular steatosis not visible by conventional light microscopy following H&E / PAS staining (page 10, lines 295-297; page 1, lines 158-160). However, there is no correlation graph shown in support of this statement with the biopsies presented in this study.
The reviewer makes a good point. In our experience, such microvesicular steatotic foci could not be predicted by H&E staining, but in most cases, these foci could only be detected in the immunohistochemical perilipin stainings. We added a new supplemental figure 4 to illustrate this phenomenon.
- Content of Table 1 is not described in a comprehensible way. What do "PLIN1/PLIN2 cluster share (%)" and "Cluster share of all hepatocytes (%)" refer to? There are huge SD values and no statistical analysis is provided in support of concluding to higher content of steatosis in biopsies of patients with HCV genotype 3.
We thank the reviewer for this feedback. We added an explanation concerning the meaning of these two parameters below the table. We have analyzed the amount of steatosis in association to the HCV genotype with Wilcoxon test. There is a significant difference in the amount of steatosis between genotype 1b and 3a (Figure 2b). There was no significant difference between the other genotypes, but notably, some of the sample sizes of the other genotype groups were quite small. Therefore, we added the sample size numbers (Figure 2b).
- Fig. 2b: By H&E staining, percentage of steatosis is higher with HCV genotype 3 chronic infection than with the majority of non-3 genotypes, which is in agreement with previously published information. Whether 1a/1b and 2a/2c are HCV recombinants or pooled subtypes is unclear. In this respect, the high steatosis rate for genotype 2 versus 2a/2c and 2b is puzzling. The number of biopsies examined in each genotype group should be indicated. A correlation graph of PLIN2 immunostaining (Fig. 2a) versus H&E staining (Fig. 2b) would objectify the conclusions
We thank the reviewer for pointing that out. Liver biopsies were taken from patients with chronic hepatitis C in the time period from 1996 to 2002 and due to the genotype testing possibilities at that time, it was not always possible to reliably determine a clear subtype, hence the designations 1a/1b and 2a/2c. Furthermore, we have added the sample sizes (Fig. 2b) to show that the sample size was sometimes variable and not representative enough in some groups to detect significant differences. In addition, we have added a correlation graph of PLIN2 immunostaining versus H&E staining (Supp. Figure 3).
- The percentages of PLIN1- and PLIN2-positive foci seem inversely correlated to the severity of steatosis (Table 2), which appears discrepant with images shown in Fig. 3b. Could the authors comment on this?
The foci were best seen in livers with only mild or moderate steatosis. In patients with high grade steatosis, nearly all hepatocytes stained positive for lipid droplets, and no foci were discernable.
- The reviewer appreciates the difficulty to perform in situ labelling of HCV proteins or RNA in chronically infected livers. However, some conclusions are provided on the basis of HCV RNA in situ hybridization (page 7, lines 226-230), whereas no RNA label is detectable in Suppl. Fig. 3. Could the authors add arrows to refer to what they identify as viral RNA in this figure?
The reviewer is right, in the compressed images uploaded for the review process, the signal of the RNA label is hardly detectable. We have uploaded the respective tiff images and have added arrows in enlarged inserts.
- Mitochondrial alterations in Fig. 4d should be pointed.
In Fig. 4d, only mild mitochondrial alterations are seen, which may not be appreciated in the given magnification. Therefore, we have included suppl. Figure 6.
- Reference to Fig. 4d for the liver zonation issue (page 10, line 283) does not seem appropriate. There is no illustration to support these conclusions.
The reviewer is correct. We refer to liver zonation demonstrated by glutamine synthetase staining in figure 4c. We have corrected the reference to the figure accordingly.
- What do the lines refer to in the non parametric Spearman correlation graphs (Suppl. Figs. 1 & 2)?
Linear regression lines and the 95 % confidence interval are illustrated in the scatter plot in order to better recognize the trend.
In conclusion, this manuscript requires profound remodeling and additional data analyses to convey clear aims and convincing conclusions with respect to the interesting question it tackles.
We have modified the manuscript accordingly and hope the manuscript may now be acceptable for publication.

Reviewer 2 Report
General Comments
Reviewed is the manuscript “Lipid droplet-associated proteins perilipin 1 and 2: molecular markers of microvesicular steatotic foci in chronic hepatitis C and other chronic liver diseases” submitted by Selina Schelbert, et, al. The authors concluded that all liver zones had steatosis foci that had mild architectural alterations but low proliferation rates, and chronic viral hepatitis commonly contains microvesicular steatotic foci. It is unclear what these foci mean clinically at this time. The material flows well throughout the discussion of each approach and the paper is properly arranged. The article is well-written, contains minimal typographical errors, and is very well-stated in terms of style and layout, but there are problems with the presentation and analysis as it is now written. After minor modifications, the item will still fulfill the criteria for publishing.
Those comments are from a statistical perspective:
• Table 1 may be made better by compiling data into the "mean (SD)" format.
• The authors indicate that P-values 0.05 (*) were regarded as significant in figure 2, whereas p 0.001 (****) was considered as significant in picture 3. Why were different significant levels chosen at different points in the manuscript?
• Both groups exhibit significant subgroup variability in Figure 3 Panels C and D. How could this be explained?
• The multiple testing correction, such as FDR, must be used due to the numerous tests that were run.
Author Response
Point to point response to Reviewer 2:
Reviewed is the manuscript “Lipid droplet-associated proteins perilipin 1 and 2: molecular markers of microvesicular steatotic foci in chronic hepatitis C and other chronic liver diseases” submitted by Selina Schelbert, et, al. The authors concluded that all liver zones had steatosis foci that had mild architectural alterations but low proliferation rates, and chronic viral hepatitis commonly contains microvesicular steatotic foci. It is unclear what these foci mean clinically at this time. The material flows well throughout the discussion of each approach and the paper is properly arranged. The article is well-written, contains minimal typographical errors, and is very well-stated in terms of style and layout, but there are problems with the presentation and analysis as it is now written. After minor modifications, the item will still fulfill the criteria for publishing.
We thank the reviewer for his general assessment of the manuscript.
Those comments are from a statistical perspective:
- Table 1 may be made better by compiling data into the "mean (SD)" format.
We have adjusted the tables accordingly.
- The authors indicate that P-values 0.05 (*) were regarded as significant in figure 2, whereas p 0.001 (****) was considered as significant in picture 3. Why were different significant levels chosen at different points in the manuscript?
We thank the reviewer for this comment. We have set up p < 0.05 as the significance level for every point, but for some correlations, the p-values were even far below the significance level. However, we adjusted the graphs with p < 0.001.
- Both groups exhibit significant subgroup variability in Figure 3 Panels C and D. How could this be explained?
- The multiple testing correction, such as FDR, must be used due to the numerous tests that were run.
We added multiple testing correction (FDR) in Figure 2b. Due to the large numbers of groups and the resulting small number of sample sizes, Kruskal-Wallis test and multiple testing correction (FDR) were not significant, although the Wilcoxon test was significant between genotype 1b and 3a. The tables 1, A1 and A2 only show mean and standard deviation calculations and are intended to be descriptive of our collective. No further statistic tests were done.
